# A type-specific B-cell epitope at the apex of outer surface protein C (OspC) of the Lyme disease spirochete, *Borreliella burgdorferi*

David J. Vance,[1,2] Grace Freeman-Gallant,[1] Kathleen McCarthy,[2] Carol Lyn Piazza,[1] Yang Chen,[3] Clint Vorauer,[4] Beatrice Muriuki,[5] Michael J. Rudolph,[3] Lisa Cavacini,[5] Miklos Guttman,[4] Nicholas J. Mantis[1]

**ABSTRACT** Broadly protective immunity to the Lyme disease spirochete, *Borreliella burgdorferi*, is constrained by antibodies against type-specific epitopes on outer surface protein C (OspC), a homodimeric helix-rich lipoprotein essential for early stages of spirochete dissemination in vertebrate hosts. However, the molecular basis for type-specific immunity has not been fully elucidated. In this report, we produced and characterized an OspC mouse monoclonal antibody, 8C1, that recognizes native and recombinant OspC type A ($OspC_A$) but not OspC type B or K. Epitope mapping by hydrogen–deuterium exchange mass spectrometry (HDX-MS) localized 8C1's epitope to a protruding ridge on the apex of $OspC_A$ α-helix 3 (residues 130–150) previously known to be an immunodominant region of the molecule. Alanine scanning pinpointed 8C1's core binding motif to a solvent exposed patch consisting of residues $K_{141}$, $H_{142}$, $T_{143}$, and $D_{144}$. Analysis of 26 Lyme disease-positive serum samples confirmed human antibody reactivity with this region of $OspC_A$, with residues $E_{140}$ and $D_{144}$ as being the most consequential. Our results underscore the importance of α-helix 3 as a target of type-specific epitopes on $OspC_A$ that should be taken into consideration in Lyme disease vaccine design.

**IMPORTANCE** A central challenge in the development of vaccines against Lyme disease, the most common vector-borne infection in the United States, is the antigenically variable nature of the lipoproteins displayed on the surface of the disease-causing spirochete, *Borreliella burgdorferi*. For example, antibodies against one type of outer surface protein C (OspC), a lipoprotein involved in *B. burgdorferi* transmission and early stages of infection, may have little or no cross reactivity with another seemingly closely related variant of OspC, thereby hampering the use of a single OspC type as a vaccine antigen. For the sake of vaccine design, it is critical to identify the specific epitopes on OspC that both restrict and enable cross-reactivity.

**KEYWORDS** vaccine, antibody, epitope, neutralizing, human

Lyme borreliosis (or Lyme disease) is a potentially debilitating tick-borne infection caused by the spirochete *Borreliella burgdorferi* sensu latu (s.l.). Following transmission via a tick bite, *B. burgdorferi* replicates locally in the skin, often presenting clinically as an expanding rash known as erythema migrans. If left untreated, the spirochete can disseminate to secondary tissues, with possible neurologic, arthritic, and cardiac complications (1, 2). In humans, *B. burgdorferi* infection is accompanied by a robust B-cell response that has been associated with disease resolution (3). *B. burgdorferi*-specific antibodies also afford protection against reinfection, albeit with the caveat that immunity is restricted to strains expressing the same outer surface protein C (OspC) type (4–11).

**Peer Reviewer** Suman Kundu, The University of Tennessee Health Science Center, Memphis, Tennessee, USA

Address correspondence to David J. Vance, david.vance@health.ny.gov, or Nicholas J. Mantis, nicholas.mantis@health.ny.gov.

The authors declare no conflict of interest.

See the funding table on p. 11.

OspC (BB_B19) is expressed by *B. burgdorferi* during tick-mediated transmission and in the early stages of mammalian infection, where it has an array of adhesin and immune evasion activities, including interactions with plasminogen and the complement component C4b (12–19). Structurally, OspC (~23 kDa) is a helical-rich polypeptide that dimerizes to form a knob-shaped molecule anchored via a lipidated N-terminus to the spirochete's outer membrane (16). In humans and other mammals, OspC is among the most immunoreactive of *B. burgdorferi*'s many outer surface proteins (20). It is also one of the most polymorphic of the spirochete's many lipoproteins, with >30 known OspC types reported worldwide (10, 21–23). Even within relatively confined geographical areas, the diversity of *ospC* alleles within tick-associated *B. burgdorferi* can be remarkably high. As a case in point, 19 different *ospC* alleles were identified within a survey of nymphal and adult ticks (*Ixodes scapularis*) from a region of high endemicity in New York State (21). Others have noted similar degrees of *ospC* diversity within colonies of wild caught ticks (24).

The polymorphic nature of *ospC* represents one of the major challenges associated with the use of OspC as a candidate Lyme vaccine (4, 6, 7, 25–27). Marconi and colleagues have overcome the challenge by generating "chimeritope" antigens consisting of concatenated epitopes (polypeptides), encompassing multiple OspC types (28). However, the "chimeritope" approach does not retain conformational tertiary and quaternary epitopes, including some associated with protection (29). An alternative approach is to preserve OspC's quaternary structure but "engineer out" immunodominant variable epitopes. This strategy is referred to as immune focusing or protein dissection and has been applied widely to other variable antigens of interest, like HIV-1's surface glycoprotein (30, 31). With that in mind, we have sought to better define the antigenic landscape of OspC and generate a high-resolution B-cell epitope map of the molecule as a basis for rational vaccine design.

## RESULTS AND DISCUSSION

Monoclonal antibodies (mAbs) are powerful tools for identifying immunodominant and subdominant B-cell epitopes on highly variable pathogen-associated surface antigens, such as SARS-CoV-2 Spike, influenza virus hemagglutinin (HA), and HIV-1 envelope glycoprotein (30, 32–34). The same approaches are being applied to OspC with the recent X-ray crystal structures of OspC bound to Fabs from mAbs B5 (PDB ID 7UIJ) and B11 (PDB ID 9BIF) (29, 35). In pursuit of additional mAbs, we immunized groups of BALB/c mice with a mixture of recombinant OspC (rOspC) types A, B and K, then screened splenic-derived B-cell hybridoma supernatants for rOspC-specific reactivity (see **Materials and Methods**). Supernatants from one particular hybridoma, 8C1, contained IgG that recognized $rOspC_A$ but not $rOspC_B$ or $rOspC_K$ by Luminex (Fig. S1A). Similarly, by dot blot, 8C1 hybridoma supernatants bound to *B. burgdorferi* strain B313 (a derivative of B31 that overexpresses $OspC_A$) but not to strains ZS7 ($OspC_B$) or 297 ($OspC_K$) (Fig. S1B). *B. burgdorferi* strains used in this study are listed in Table 1.

To further characterize 8C1, the hybridoma was single cell cloned, and the $V_H$ and $V_L$ coding regions were amplified from hybridoma-derived cDNA, subjected to DNA sequencing, and subsequently cloned as gBlocks into pcDNA3.1-based human $IgG_1$ Fc and kappa light chain expression plasmids. Recombinant 8C1 IgG1 had an rOspC reactivity profile identical to the hybridoma-derived 8C1, indicating successful reconstitution of the $V_H$ and $V_L$ pairing (Fig. 1; Fig. S2). By flow cytometry, 8C1 bound to *B. burgdorferi* strain B313 ($OspC_A$) but not a *B. burgdorferi ospC* deletion mutant (B31 A3 Δ*ospC*) or strains displaying $OspC_B$ (ZS7) or $OspC_K$ (297) (Fig. 1A). In addition, 8C1 induced both agglutination and alterations in outer membrane integrity of *B. burgdorferi* B313, a derivative of B31 that overexpresses OspC due to a genomic deletion of *ospAB* (Fig. 1B and C)(41). We have speculated that the ability of mAbs to promote spirochete agglutination and PI sensitivity may reflect their ability to impact *B. burgdorferi* migration and transmission (35, 42). It should be noted that 8C1's capacity to agglutinate strain B313 (~15%) is notably lower than reported for B5 or B11 (~30% each), possibly due

**TABLE 1** *B. burgdorferi* strains used in this study

| Strain | *ospC* allele | Source | Reference |
|---|---|---|---|
| B31 | ospC$_A$ | ATCC | (36) |
| B31-A3[a] | ospC$_A$ | Dr. Yi-Pin Lin | (37) |
| B31 A3 ΔospC[b] | N/A[e] | Dr. Yi-Pin Lin | (38) |
| ZS7 | ospC$_B$ | Dr. Yi-Pin Lin | (39) |
| 297 | ospC$_K$ | ATCC | (40) |
| B313[c] | ospC$_A$ | Dr. Yi-Pin Lin | (41) |
| GGW941[d] | ospC$_A$ | In house | (35) |

[a]Low passage, infectious derivative of B31 that lacks cp9.
[b]Derivative of B31 (also referred to as "B31-A3OspCK1") that lacks the entire ospC coding sequence due to insertion cassette.
[c]A clonal, non-infectious, high passage derivative of B31 lacking *ospAB* that, consequently, expresses high levels of OspC.
[d]A derivative of B31 that expresses rpoS ectopically under control of an IPTG-inducible promoter.
[e] N/A, not applicable.

to differences in epitope specificity and/or binding affinity (35). To address the issue of binding affinity, 8C1 was subjected to bio-layer interferometry (BLI) to determine binding kinetics to rOspC$_A$. By using this method, the estimated dissociation constant (K$_D$) of 8C1 Fab fragments (monovalent interactions) for rOspC$_A$ was 10.7 nM, and 8C1 IgG for rOspC$_A$ (bivalent interactions) was 11.2 nM (Fig. S2B and C).

We next investigated whether 8C1 induces complement-dependent and/or -independent motility arrest of *B. burgdorferi*. Motility is critical for spirochete migration during tick transmission and within skin tissues (43). To circumvent issues associated with intrinsically low OspC expression by *B. burgdorferi* B31 in culture, we utilized a recently constructed strain with an IPTG inducible *rpoS* allele, thereby activating native *ospC* expression *in trans* (35). In the absence of complement, *B. burgdorferi* cells treated with ≥10 µg/mL of 8C1 IgG were significantly less motile than isotype controls (Table

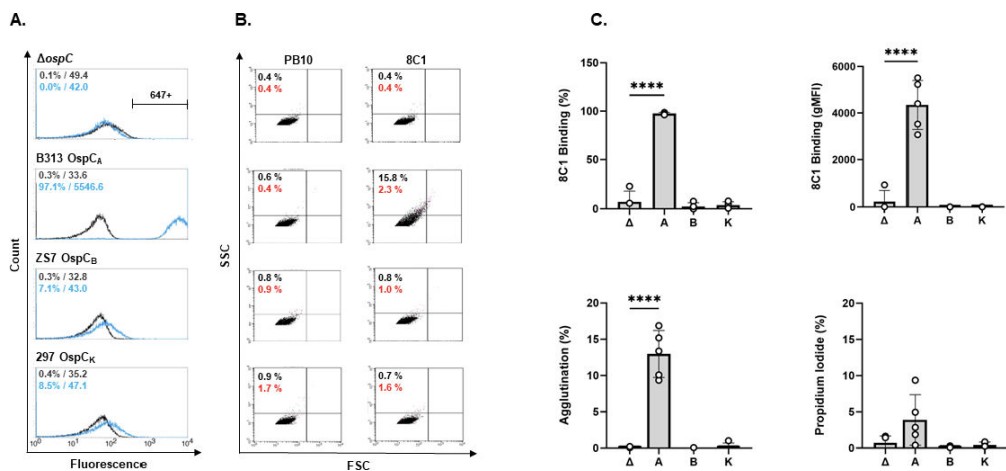

**FIG 1** 8C1 recognizes native OspC on the surface of *B. burgdorferi*. Cultures of B. *burgdorferi* B31A3 Δ*ospC*, B313, ZS7, and 297 (as described in the text and Table 1) were probed with 8C1 hIgG or an isotype control (PB10 hIgG), then stained with an Alexa Fluor 647-labeled anti-human IgG secondary antibody and subjected to flow cytometric analysis. (A) Representative histograms of 8C1 IgG (blue) and PB10 IgG (black) with text insets indicating % positive cells and gMFI for each strain examined. (B) Corresponding FSC (*x*-axis) and SSC (*y*-axis) dot plots representing event size and granularity, respectively. The percent of agglutinated events (black text insets) was calculated from the sum of UL + UR + LR quadrants, relative to the total events counted (20,000). The percent of events positive for propidium iodide (red text insets) are shown as red dots. The anti-ricin humanized mAb PB10 was used as an isotype control (C) Combined flow cytometry analysis from *n* ≥ 4 independent biological replicates, with each symbol representing an independent experiment, the column representing the mean, and the error bars representing standard deviations. The OspC type (null, A, B, and K) is plotted on the *x*-axis. The gMFI from PB10 considered as background was subtracted from values plotted. Asterisks specify statistical significance (*P* < 0.0001; one-way ANOVA).

2). The addition of 20% human complement further reduced the motility of 8C1-treated cells, although the difference did not achieve statistical significance, indicating that the majority of 8C1's effects on motility are complement-independent. This contrasts with mAb B5, which when evaluated in parallel had demonstrable complement-dependent borreliacidal activity (Table 2). The relevance of these observations to *in vivo* activity remains to be determined, as the exact concentrations and mechanism(s) by which OspC antibodies interfere with *B. burgdorferi* transmission and infection remain incompletely defined (6, 44, 45).

We next sought to localize 8C1's epitope on $OspC_A$. In competitive binding assays, neither B5 nor B11 inhibited 8C1 from associating with $rOspC_A$ (data not shown), indicating that 8C1's epitope is unlikely to be situated on the lateral face of OspC (29, 35). Therefore, we turned to hydrogen–deuterium exchange mass spectrometry (HDX-MS) to identify regions of OspC that interact with 8C1 (46). A series of preliminary quench and digestion experiments revealed that proteolysis with nepenthesin II without addition of urea generated the largest set of observable peptides for $OspC_A$. After filtering out weak and overlapping signals, there were 73 unique peptides remaining, resulting in a final sequence coverage of 98.8% with a redundancy of 5.3. The HDX-MS profiles of $rOspC_A$ without (unliganded) and with twofold molar excess of mAb 8C1 were compared. The magnitude of changes across $rOspC_A$ in the presence of 8C1 was minor, with only a few regions showing a statistically significant degree of protection. Among these few regions of $OspC_A$ were peptides spanning residues 88–98 and 134–144, which correspond to the apex of the $OspC_A$ dimer (Fig. 2A; File S1). Minor protection was also detected along distal residues 145–156 but not the proximal peptide spanning residues 134–141. Based on this profile, we speculated that 8C1's epitope is centered around $OspC_A$ residues $H_{142}$, $T_{143}$, and $D_{144}$.

Residues 142–144 are nested within several previously reported mouse and human linear B-cell epitopes, including the borreliacidal mAb, 16.22 (Table 3) (25, 47–49). In fact, Marconi and colleagues refer to this region of OspC as loop 5 (25, 47). We therefore evaluated 8C1 reactivity with OspC-derived peptides, spanning residues 130–150. In a microsphere immunoassay (MIA), 8C1 reacted strongly to $OspC_A$ peptides 130–150 (Fig. 2C) and to a lesser degree with peptides 132–146 (data not shown). By BLI, 8C1 Fabs had an affinity constant ($K_D$) of 9.2 nM for peptides 130–150 compared with 10.7 nM for recombinant, dimeric $rOspC_A$, demonstrating that a linear epitope may account for a large proportion of 8C1's binding energetics (Fig. S3). To define the critical residues associated with 8C1 interactions with the peptide, nine surface-exposed residues within amino acids 130–150 (Fig. 2B) were subjected to alanine (Ala) mutagenesis, and the resulting biotinylated peptides were probed with 8C1. Ala substitutions at residues $K_{141}$, $H_{142}$, $T_{143}$, and $D_{144}$ each reduced 8C1 binding by 100- to 10,000-fold relative to native peptides 130–150 (Fig. 2C). These results are consistent with the HDX-MS analysis and demonstrate that $K_{141}$, $H_{142}$, $T_{143}$, and $D_{144}$ likely constitute important 8C1 contact points on $OspC_A$. Moreover, those four residues alone may account for 8C1's restricted

**TABLE 2** Antibody-mediated motility arrest (%) of *B. burgdorferi* GGW941

| mAr[a] | 8 C1 | | B5 | | PB10 | |
|---|---|---|---|---|---|---|
| | −[b] | + | − | + | − | + |
| 0 | 100 ± 0 | 71 ± 12 | 100 ± 0 | 71 ± 12 | 100 ± 0 | 71 ± 12 |
| 1 | 73 ± 19 | 63 ± 27 | 65 ± 15 | 62 ± 17 | 85 ± 14 | 83 ± 2 |
| 3 | 60 ± 14 | 54 ± 24 | 40 ± 7** | 17 ± 14** | 74 ± 20 | 69 ± 14 |
| 10 | 47 ± 24* | 35 ± 7* | 20 ± 8** | 17 ± 6** | 81 ± 10 | 67 ± 16 |
| 20 | 42 ± 1* | 45 ± 27* | 4 ± 4** | 7 ± 6** | 71 ± 12 | 73 ± 5 |
| 30 | 40 ± 13* | 51 ± 27* | 8 ± 7** | 7 ± 2** | 79 ± 11 | 68 ± 10 |

[a]mg/mL.
[b]indicates without (-) or with (+) addition of human complement. Data points that are significantly different from the isotype control (PB10 IgG1) are indicated with asterisks i (*<0.01; ** <0.001). Statistical analysis was determined using two-way ANOVA with Dunnett's multiple comparisons test in which experimental means (8C1, B5) were compared to mean of the control (PB10) at each concentration.

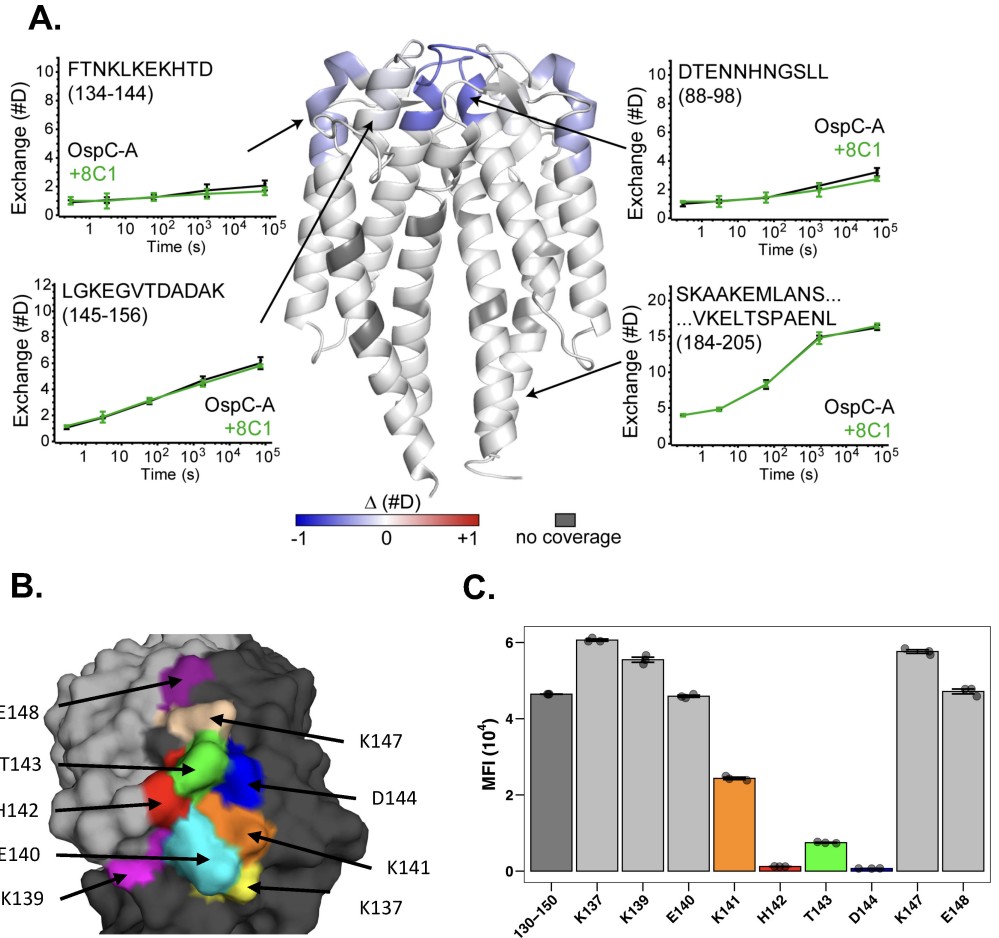

FIG 2 Localization of 8C1's epitope to OspC$_A$ residues 141–144. (A) ΔHDX in rOspC$_A$ upon incubation with MAb 8C1 is plotted on the structure of OspC$_A$ (PDB ID 9BIF) in the center of the panel. Regions with increased protection are colored in blue and more exposed in red. Deuterium uptake plots for unbound rOspC$_A$ (black) and 8C1–rOspC$_A$ complex (green) are shown in plots for selected regions indicated by arrows. Symbols represent the mean ± SD from three independent measurements. (B) Surface representation of dimeric OspC$_A$ (PDB ID 1GGQ; monomers colored gray and black) with surface exposed residues targeted for alanine mutation colored by residue and labeled. (C) 8C1 reactivity (MFI) with OspC$_A$ peptide 130–150 (left column, dark gray) or peptides 130–150 with Ala substitutions as indicated on the x-axis. The colored columns are color coded based on Panel B.

OspC reactivity, as an alignment of 23 OspC types revealed that only Types A and C contain the $K_{141}$, $H_{142}$, $T_{143}$, and $D_{144}$ motif (Table 4). In fact, the presence of a native Ala at position 143 in 16 OspC types rather than OspC$_A$ $T_{143}$ is theoretically enough to completely abrogate 8C1 reactivity.

We recently extended the work by Buckles and colleagues that OspC types A-, B-, and K-derived peptides encompassing residues 132–146 are recognized by sera from individuals positive for Lyme disease (47, 48). Therefore, to investigate the contribution of the $K_{141}$, $H_{142}$, $T_{143}$, and $D_{144}$ motifs in the peptide recognition in humans, OspC$_A$ peptides 130–150 and the Ala mutants were probed with *B. burgdorferi* seropositive serum samples (n = 26 total). Specifically, we employed human serum samples provided by the Lyme Disease Biobank at Nuvance Health (Danbury, CT) that had been deemed positive in a standard or modified two-tiered Lyme disease test (48). As samples were principally from individuals with post-treatment Lyme disease (PTLD), only IgG reactivity was examined.

Peptide recognition by virtually every human serum sample was negatively affected by Ala substitutions at residues $E_{140}$ or $D_{144}$ (Fig. 3). Ala substitutions elsewhere in the

**TABLE 3**  Reported OspC linear B-cell epitopes overlapping with 8C1

| IEDB ID | Residues | OspC type | Species | Reference |
|---------|----------|-----------|---------|-----------|
| 6984 | 130–150 | A | Human, Mouse | (47) |
| 57644 | 131–149 | A | Mouse | (50) |
| 63756 | 133–147 | A | Mouse | (49) |
| 745113 | 132–144 | K | Human | (51) |
| 745097 | 133–145 | M | Human | (51) |

peptide enhanced antibody binding, a phenomenon observed by others for reasons possibly relating to epitope unmasking (52, 53). These findings confirm that this region is highly immunogenic in people with Lyme disease and suggests that there are linear epitopes antigenic in humans that are dependent on $E_{140}$ and $D_{144}$.

## Conclusions

OspC defines strain-specific immunity to *B. burgdorferi* and therefore limits its utility as a vaccine (4). However, the B-cell epitopes that constrain antibody reactivity to one OspC type at the expense of others are not fully characterized. In this report, we defined a linear epitope situated at the apex of $OspC_A$ necessary for recognition by the mAb 8C1. The core of 8C1's epitope consists of residues $K_{141}$, $H_{142}$, $T_{143}$, and $D_{144}$, which constitutes a sequence variable but structurally conserved region on α-helix 3. Indeed, in our effort to better define the B-cell epitopes on OspC, we inadvertently "rediscovered" an epitope hotspot (25, 49, 51). For example, Yang and colleagues identified this region as the target of the complement-independent, borreliacidal mouse mAb 16.22 (49). Peptide reactivity profiles led those investigators to conclude that 16.22 recognizes the core $K_{139}$–$E_{140}$–$K_{141}$ motif. However, based on their results, it is equally plausible that

**TABLE 4**  Alignments of OspC residues 141–144

| Type[a] | 141 | 142 | 143 | 144 | GenBank ID |
|---------|-----|-----|-----|-----|------------|
| A | **K**[b] | **H** | **T** | **D** | X69596 |
| A3 | E | Q | A | T | EF592541 |
| B | N | **H** | A | Q | CP001422 |
| C | **K** | **H** | **T** | **D** | DQ437462 |
| C3 | S | **H** | A | E | EF592543 |
| D | N | Q | A | E | CP001484 |
| D3 | S | **H** | A | V | EF592544 |
| E | E | **H** | A | V | AY275221 |
| E3 | S | **H** | G | E | EF592545 |
| F | G | N | A | Q | L42896 |
| F3 | E | **H** | **T** | **D** | EF592547 |
| G | S | N | A | **D** | AY275223 |
| H | E | **H** | A | S | CP001271 |
| H3 | S | **H** | G | N | FJ932733 |
| I | E | **H** | **T** | **D** | AY275219 |
| I3 | G | N | A | Q | FJ932734 |
| J | S | **H** | A | E | CP001535 |
| K | E | **H** | A | Q | AY275214 |
| L | E | N | V | A | EU375832 |
| M | S | **H** | A | E | CP001550 |
| N | S | **H** | A | Q | EU377775 |
| T | G | **H** | A | E | AY275222 |
| U | S | **H** | A | **D** | CP001493 |

[a]OspC types.
[b]**Bold** font indicates positions of identity relative to Type A.

## A. Individual Lyme disease serum samples (n=26)

| # | K137 | K139 | E140 | K141 | H142 | T143 | D144 | K147 | E148 |
|---|------|------|------|------|------|------|------|------|------|
| 1 | 2.80 | 1.85 | 0.53 | 1.76 | 2.19 | 1.55 | 0.77 | 2.25 | 1.19 |
| 2 | 1.67 | 1.25 | 0.54 | 1.64 | 1.56 | 1.12 | 0.67 | 1.52 | 1.38 |
| 3 | 0.87 | 0.74 | 0.54 | 0.99 | 0.91 | 0.76 | 0.76 | 0.86 | 0.69 |
| 4 | 2.93 | 1.87 | 0.56 | 1.92 | 2.48 | 1.68 | 0.69 | 2.58 | 1.28 |
| 5 | 1.95 | 1.59 | 0.57 | 1.46 | 1.73 | 1.23 | 0.62 | 1.74 | 0.92 |
| 6 | 1.84 | 1.47 | 0.59 | 1.48 | 1.57 | 1.19 | 0.70 | 1.66 | 0.90 |
| 7 | 2.58 | 1.79 | 0.63 | 1.79 | 2.17 | 1.62 | 0.71 | 2.18 | 1.19 |
| 8 | 2.21 | 1.57 | 0.64 | 1.63 | 1.94 | 1.50 | 0.80 | 1.90 | 1.15 |
| 9 | 2.16 | 1.69 | 0.64 | 1.91 | 1.99 | 1.46 | 0.78 | 1.84 | 0.98 |
| 10 | 2.74 | 1.79 | 0.64 | 1.91 | 2.16 | 1.63 | 0.73 | 2.34 | 1.27 |
| 11 | 2.29 | 1.71 | 0.65 | 1.91 | 1.96 | 1.47 | 0.78 | 2.01 | 1.11 |
| 12 | 3.18 | 1.90 | 0.65 | 1.95 | 2.73 | 1.84 | 0.75 | 2.67 | 1.48 |
| 13 | 2.34 | 1.80 | 0.66 | 1.91 | 2.08 | 1.41 | 0.79 | 1.98 | 1.07 |
| 14 | 1.51 | 1.43 | 0.67 | 2.69 | 1.18 | 0.99 | 1.58 | 1.10 | 1.09 |
| 15 | 2.37 | 1.57 | 0.68 | 1.73 | 2.26 | 1.49 | 0.86 | 2.12 | 1.18 |
| 16 | 2.51 | 1.64 | 0.68 | 1.78 | 2.15 | 1.64 | 0.87 | 2.24 | 1.28 |
| 17 | 4.58 | 2.29 | 0.68 | 2.37 | 3.69 | 2.36 | 0.77 | 3.73 | 1.54 |
| 18 | 2.16 | 1.56 | 0.73 | 1.82 | 1.85 | 1.44 | 0.91 | 1.90 | 1.07 |
| 19 | 2.04 | 1.56 | 0.75 | 1.83 | 1.83 | 1.42 | 0.88 | 1.83 | 1.14 |
| 20 | 2.38 | 1.59 | 0.77 | 1.57 | 1.71 | 1.42 | 0.70 | 1.93 | 1.09 |
| 21 | 1.34 | 0.54 | 0.80 | 1.35 | 0.86 | 0.91 | 0.84 | 1.28 | 0.86 |
| 22 | 1.05 | 1.64 | 0.86 | 2.44 | 1.65 | 1.20 | 1.12 | 1.72 | 1.11 |
| 23 | 1.29 | 1.06 | 0.91 | 0.78 | 0.70 | 1.12 | 0.53 | 1.48 | 0.96 |
| 24 | 1.58 | 2.01 | 1.22 | 1.19 | 0.69 | 1.58 | 0.80 | 2.08 | 1.27 |
| 25 | 4.08 | 5.51 | 1.38 | 6.17 | 4.71 | 2.87 | 2.00 | 3.99 | 2.16 |
| 26 | 2.38 | 1.41 | 1.45 | 0.64 | 0.36 | 1.69 | 0.21 | 2.25 | 1.49 |

## B. Impact (mean % ± SD) of Ala substitution on Ab binding

|       | K137 | K139 | E140 | K141 | H142 | T143 | D144 | K147 | E148 |
|-------|------|------|------|------|------|------|------|------|------|
| M     | 2.26 | 1.72 | 0.75 | 1.87 | 1.89 | 1.48 | 0.83 | 2.05 | 1.19 |
| ±SD   | 0.82 | 0.84 | 0.24 | 0.97 | 0.89 | 0.42 | 0.32 | 0.67 | 0.28 |

**FIG 3** Recognition of OspC$_A$ residues 137–150 by *B. burgdorferi* seropositive serum samples. (A) Lyme disease-positive human serum samples ($n = 26$) previously determined to react with OspC$_A$ residues 130–150 were subjected to MIA using 130–150 peptides with indicated single Ala substitutions, as described in the Material and Methods. The value in each box represents percent binding relative to the wild-type peptide. Values <1 (colored red) indicate a reduction in peptide recognition. (B) Mean ± standard deviation of MFI ratios for each Ala substitution for all 26 serum samples.

16.22 recognizes the KHTD motif just like 8C1. Either way, 16.22 and 8C1 most certainly have overlapping binding sites on OspC$_A$ and share the capacity to induce *B. burgdorferi* motility and growth arrest.

Even before 16.22 was reported, Marconi and colleagues had identified residues 135–150 as being an immunodominant element on OspC, a region they referred to as loop 5, in mice and humans (25, 47). Indeed, they made the case that loop 5 is highly conserved within a given OspC type but variable across types. For example, 53 of 57 (>90%) OspC$_A$ sequences were identical across this region. Our results are consistent with that observation, with human immune sera being particularly prone to interactions with E$_{140}$ and D$_{144}$. More recently, Tokarz and colleagues demonstrated that antibody

reactivity to analogous regions on $OspC_K$ (residues 132–144) and $OspC_M$ (residues 133–145) is diagnostic for Lyme disease (51). Thus, our report underscores the importance of the so-called loop 5 as a location of immunodominant and type-restricted epitopes on OspC in mice and humans with implications for both Lyme diagnostics and vaccine design.

## MATERIALS AND METHODS

### OspC protein expression

Recombinant *B. burgdorferi* $rOspC_A$ (residues 38 to 201; PDB ID 1GGQ; UniProt ID Q07337) (54), $rOspC_B$ (residues 38 to 202; *B. burgdorferi* strain ZS7; PDB ID 7UJ2) (55), and $rOspC_K$ (residues 38–202; *B. burgdorferi* strain 297; PDB ID 7UJ6) (40) were expressed in *Escherichia coli* strain BL21 (DE3) and purified by nickel-affinity and size-exclusion chromatography, as described (29).

### Bacterial strains and culture conditions

*B. burgdorferi* strains used in this study are listed in Table 1. *B. burgdorferi* strains expressing OspC types A (B313), B (ZS7), K (297), and the *ospC* deletion strain B31A3ΔospC were cultured in BSK-II medium at 37°C with 5% $CO_2$ to mid-log phase (56). Base BSK-II medium was prepared by the Wadsworth Center's Tissue and Media Core Facility and filter sterilized (0.2 µm) prior to use. *B. burgdorferi* cultures were routinely inspected for culture viability and motility during *in vitro* culture maintenance prior to the initiation of any experiments.

### Dot blot

Bacterial strains expressing OspC types A (B313), B (ZS7), K (297), and deletion strain B31A3ΔospC were cultured in BSK-II medium at 37°C with 5% $CO_2$ to mid-log phase, collected by centrifugation (3,300×*g*), washed with PBS, and stored at −20°C until needed. The bacterial pellets and aliquots of rOspC types A, B, and K were diluted 10-fold in PBS before spotting on nitrocellulose membrane. PBS and an unrelated protein, rOspA (strain B31), were included as negative controls. Incubation with 8C1 hybridoma supernatant and processing and analysis of the dot blot were performed as described (42).

### Surface labeling and antibody-mediated agglutination of *B. burgdorferi*

*B. burgdorferi* strains expressing OspC types A (B313), B (ZS7), K (297), and B31A3 Δ*ospC* were treated with 8C1 human IgG1 (10 µg/mL) and analyzed by flow cytometry, as recently described (29). The anti-ricin mAb PB10 IgG1 was used as an isotype control (57). Briefly, strains were cultured in BSK-II media minus gelatin at 37°C with 5% $CO_2$ to mid-log phase. Cells were collected by centrifugation (3,300×*g*), washed with PBS, resuspended in media minus phenol red, and incubated at room temperature for 30 min. A total of $5 \times 10^6$ cells in 50 µL volume were incubated with 8C1. In surface-bound antibodies, the cells were detected with goat anti-human IgG (H + L) cross-adsorbed secondary antibody Alexa Fluor 647 (Invitrogen). Propidium iodide (0.75 µM; Sigma) was added to the culture for detection of membrane permeability. Analysis was conducted on a BD FACSCalibur flow cytometer (BD Biosciences Franklin Lakes, NJ). Cells were gated on forward scatter and side scatter to assess aggregate size and granularity. Alexa Fluor 647 fluorescence, PI staining, and agglutination were measured on 20,000 events using CellQuest Pro software (BD Biosciences, Franklin Lakes, NJ). Agglutination was calculated by the sum of events in the upper-left, upper-right, and lower-right quadrants relative to total events counted, as reported (58).

## Mouse immunizations and B cell hybridoma production

Animal studies were approved by the Wadsworth Center's Institutional Animal Care and Use Committee (IACUC). Female BALB/c mice of ~10 weeks of age were immunized with a combination of rOspC Types A, B, and K (20 µg each; 60 µg total) in 100 µL via the intraperitoneal route. The proteins were emulsified in 50% TiterMax Gold adjuvant (Sigma Aldrich, St Louis, MO). Mice were immunized three times at 3 week intervals, and rOspC-specific titers were confirmed by indirect enzyme-linked immunosorbent assay (ELISA). Mice were boosted with rOspC (without adjuvant) 4 days before being sacrificed. Splenocytes were mixed 1:2 with Sp2/0 mouse myeloma cells and fused with PEG and subject to hypoxanthine–aminopterin–thymidine (HAT) selection (59, 60). Supernatants were tested by multiplexed immunoassay (MIA) for reactivity with rOspC types A, B, and K (see below). Positive B-cell hybridomas wells were cloned by two rounds of single cell dilution then and transitioned into hypoxanthine–thymidine (HT) medium.

## 8C1 $V_H$ and $V_L$ sequence determinations and IgG expression

The murine $V_H$ and $V_L$ cDNA sequences were determined as described (61). Briefly, total RNA was extracted from the 8C1 hybridomas via the RNeasy Mini Kit (Qiagen), reverse transcribed into cDNA using the Smartscribe Reverse Transcriptase Kit (Takara Bio, San Jose, CA), and then used as template for PCR with reverse primers for the constant regions of the heavy, light kappa, and light lambda genes, as well as the universal forward primer added during reverse transcription. PCR products were cloned into the Zero Blunt TOPO vector (Thermo Fisher Scientific) and transformed into Top Ten *E. coli* (Thermo Fisher Scientific). Several individual colonies from each transformation were picked, grown overnight, miniprepped to obtain the plasmid, and submitted for Sanger sequencing using the forward and reverse primers supplied in the Zero Blunt cloning kit. The mouse immunoglobulin heavy- and light-chain genes (VH/VL) were cloned into pcDNA3.1 in-frame with human IgG1 and human kappa chain backbone (Genscript, New Jersey). Equal amounts of heavy- and light-chain plasmids were transfected into Expi293 cells using ExpiFectamine293 transfection reagents (Thermo Scientific, Waltham, MA), following manufacturer's instructions. Supernatants containing the secreted antibodies were harvested, clarified, and antibody purified using protein A chromatography. The purified antibodies were buffer exchanged in PBS and stored at 4°C.

## Biolayer interferometry (BLI)

8C1 affinity was measured on an Octet RED96e Biolayer Interferometer (Sartorius, Goettingen, Germany) using the Data Acquisition 12.0 software. rOspC$_A$ was biotinylated using EZ-link NHS-biotin kit (Thermo Fisher Scientific). Biotinylated 130–150 peptides were received from Genemed (San Antonio, TX). Biotinylated rOspC$_A$ (3 µg/mL) or biotinylated peptide (0.25 µg/mL) in PBS containing 2% w/v BSA (buffer) was captured onto Octet SA (streptavidin) biosensors (Sartorius) for 5 min. After 3 min to equilibrate to baseline in buffer, sensors were then immersed in a twofold dilution series of 8C1 mAb or Fab, starting at 100 nM, for 10 min. The sensors were then dipped into wells containing buffer for 30 min to allow for dissociation. The raw sensor data were loaded into the Data Analysis HT 12.0 software, grouped and fit using a 1:2 bivalent analyte model (mAb) or a 1:1 model (Fab).

## *B. burgdorferi* motility determinations by dark-field microscopy

Detailed strain description and methods associated with *B. burgdorferi* motility assays are described elsewhere (62). Briefly, mid-log-phase cultures of a *B. burgdorferi* B31 derivative (GGW941) carrying an IPTG-inducible *rpoS* allele was treated with OspC mAbs (1–30 µg/mL) in the absence or presence of 20% human complement (Sigma-Aldrich) for 16 or 24 h. After which, cultures were examined in a double-blind fashion by dark-field microscopy for motile spirochetes using a Trinocular DF microscope (AmScope) equipped with a camera with reduction lens (AmScope SKU: MU1603) and

a 40– dry dark-field condenser (AmScope; DK-DRY200). Spirochetes were considered nonviable when complete loss of motility and refractivity was observed. Spirochetes were enumerated in at least four visual fields, and the percent viability was calculated as the ratio of live spirochetes (mean of four fields) in treated samples to spirochetes in the untreated control samples (mean of four fields). Polyclonal serum from *B. burgdorferi*-infected mice and mAb B5 were used as positive controls; naive serum and the anti-ricin mAb, PB10, were used as negative controls. This experimental set up was conducted over the course of three independent sessions, and data are plotted as the means for the 3 days of counting. Statistical analysis was determined using two-way ANOVA with Dunnett's multiple comparisons test in which experimental means (8C1, B5) were compared to control mean (PB10) for each antibody concentration.

## Microsphere immunoassay (MIA)

rOspC$_A$, rOspC$_B$, and rOspC$_K$ were coupled to MagPlex-C microspheres (5 µg antigen/1 × 10$^6$ microspheres) via the xMAP Antibody Coupling Kit (Luminex Corporation, Austin, TX) as recommended by the manufacturer. To couple biotinylated peptides (Genemed, San Antonio, TX), MagPlex-Avidin microspheres were suspended in assay buffer (1× PBS, 2% BSA, pH 7.4) with biotinylated peptides (5 µg/1 × 10$^6$ microspheres) and allowed to incubate in a tabletop rotator at room temperature for 30 mins. Avidin microspheres were washed three times using a magnetic separator and wash buffer (1× PBS, 2% BSA, 0.02% TWEEN-20, 0.05% Sodium azide, pH 7.4), resuspended in assay buffer, and stored at 2°C– 8°C until use.

Coupled microspheres were diluted in assay buffer (1:50) and then added (50 µL) to black, clear-bottomed, non-binding, chimney 96-well plates (Greiner Bio-One, Monroe, North Carolina). For the initial screens of hybridoma supernatant, 50 µL (neat) was added to each well. For the alanine mutant scan assays, 8C1 was diluted to 5 µg/mL. Human serum samples positive in a standard or modified two-tiered Lyme disease test, kindly provided by the Lyme Disease Biobank at Nuvance Health (Danbury, CT), were diluted 1:100 in assay buffer and then added (50 µL) to each well (48). Plates were incubated for 1 h in a tabletop shaker (600 rpm) at room temperature and then washed three times using a magnetic separator and wash buffer. For the hybridoma supernatant screens, goat anti-mouse IgG, Human-ads-PE (SouthernBiotech, Birmingham, Alabama) secondary antibody was diluted 1:500 in assay buffer and added (100 µL) to each well. For the alanine mutant scan assays, PE-labeled goat anti-Human IgG Fc, eBioscience (Invitrogen, Carlsbad, California) was diluted 1:500 in assay buffer and added (100 µL) to each well. Secondaries were incubated at room temperature for 30 min in a tabletop shaker (600 rpm). Plates were washed as previously stated, resuspended in 100 µL of wash buffer, and analyzed using a FlexMap 3D (Luminex Corporation). Both assay and wash buffers were prepared by the Wadsworth Center's Cell and Tissue Culture core facility.

## HDX-MS

Stock concentrations of rOspC$_A$ (8.5 µM) in PBS either alone or in a complex with a twofold excess of antibody were diluted into 90 µL of deuterated PBS buffer (20 mM phosphate, 150 mM NaCl, 0.02% sodium azide, 1 mM EDTA pH* 7.54, 85%D final) containing 0.2 nM bradykinin and incubated 3 s on ice, or either 3 s, 1 minute, 30 min, or 20 h at 21°C. Each starting stock also included a mixture of imidazolium compounds to serve as exchange reference standards (63). At the desired time point, the sample was rapidly mixed with an equal volume of ice cold 0.2% formic acid and 0.1% trifluoroacetic acid (TFA) for a final pH of 2.5. Samples were then immediately frozen on ethanol/dry ice and stored at −80°C until LC-MS analysis. Undeuterated samples were prepared the same way but with undeuterated buffer for each step.

Samples were thawed at 5°C for 8 min and injected using a custom LEAP robot integrated with an LC-MS system (64). The protein was first passed over a Nepenthesin II column (2.1 × 30 mm; AffiPro) at 400 µL/min for inline digestion with the protease

column held at 20°C. Peptides were then trapped on a Waters XSelect CSH C18 trap cartridge column (2.1 × 5 mm, 2.5 µm) and resolved over a CSH C18 column (1 × 50 mm, 1.7 µm, 130 Å) using linear gradient of 5 to 35% B (A: 0.1% FA, 0.025% TFA, 5% ACN; B: ACN with 0.1% FA) over 10 min and analyzed on a Thermo Orbitrap Ascend mass spectrometer at a resolution setting of 120,000. A series of washes over the trap and pepsin columns was used between injections to minimize carry-over as described (64). Data dependent MS/MS acquisition was performed on an undeuterated sample using rapid CID and HCD scans and processed in Byonic (Protein Metrics) with a score cutoff of 150 to identify peptides. Deuterium incorporation was analyzed using HDExaminer v3 (Trajan Scientific and Medical) (65).

## ACKNOWLEDGMENTS

We gratefully acknowledge Mrs. Elizabeth Cavosie for administrative assistance. We thank the Applied Genomic Technologies core for DNA sequencing services, the Immunology core for access to flow cytometer, and the Media and Tissue Culture core for bacterial media. We gratefully acknowledge Dr. John Martignetti and Lisa Arrigo of the Nuvance Health Lyme disease Biobank for generously providing the Lyme disease serum samples used in this study.

This work was supported by the National Institute of Allergy and Infectious Diseases (NIAID), National Institutes of Health, Department of Health and Human Services, Contracts No. 75N93019C00040 and 75N93024C00069 (PI/PD Mantis). HDX instrumentation at the University of Washington was supported by award S10OD030237 from the National Institute of General Medical Sciences (NIGMS). The content is solely the responsibility of the authors and does not necessarily represent the official views of the National Institutes of Health.

## AUTHOR AFFILIATIONS

[1]Division of Infectious Diseases, Wadsworth Center, New York State Department of Health, Albany, New York, USA
[2]Department of Biomedical Sciences, University at Albany, Albany, New York, USA
[3]New York Structural Biology Center, New York, New York, USA
[4]Department of Medicinal Chemistry, University of Washington, Seattle, Washington, USA
[5]University of Massachusetts Chan Medical School, Worcester, Massachusetts, USA

## AUTHOR ORCIDs

David J. Vance  http://orcid.org/0000-0002-9913-7722
Grace Freeman-Gallant  http://orcid.org/0009-0003-7154-2763
Carol Lyn Piazza  http://orcid.org/0009-0004-6482-3871
Lisa Cavacini  http://orcid.org/0000-0003-1417-8339
Miklos Guttman  http://orcid.org/0000-0003-2419-1334
Nicholas J. Mantis  http://orcid.org/0000-0002-5083-8640

## FUNDING

| Funder | Grant(s) | Author(s) |
| --- | --- | --- |
| HHS | NIH | National Institute of Allergy and Infectious Diseases (NIAID) | 75N93019C00040, 75N93024C00069 | Nicholas J. Mantis |
| HHS | NIH | National Institute of General Medical Sciences (NIGMS) | S10OD030237 | Miklos Guttman |

## AUTHOR CONTRIBUTIONS

David J. Vance, Formal analysis, Investigation, Methodology, Resources, Writing – original draft, Writing – review and editing | Grace Freeman-Gallant, Investigation, Methodology, Writing – original draft | Kathleen McCarthy, Investigation | Carol Lyn Piazza, Investigation | Yang Chen, Investigation | Clint Vorauer, Investigation | Beatrice Muriuki, Investigation | Michael J. Rudolph, Investigation, Resources | Lisa Cavacini, Investigation, Project administration, Resources | Miklos Guttman, Formal analysis, Investigation, Methodology, Project administration, Supervision, Writing – original draft | Nicholas J. Mantis, Conceptualization, Formal analysis, Funding acquisition, Investigation, Project administration, Supervision, Writing – original draft, Writing – review and editing

## ADDITIONAL FILES

The following material is available online.

### Supplemental Material

**Supplemental figures (Spectrum02883-24-s0001.pdf).** Fig. S1 to S3.

### Open Peer Review

**PEER REVIEW HISTORY (review-history.pdf).** An accounting of the reviewer comments and feedback.

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
