## [Reviewer comments · Microbiology Spectrum]

Microbiology Spectrum

A type-specific B cell epitope at the apex of Outer surface protein C (OspC) of the Lyme disease spirochete, *Borrelia burgdorferi*

David Vance, Grace Freeman-Gallant, Kathleen McCarthy, Carol Lyn Piazza, Yang Chen, Clint Vorauer, Beatrice Muriuki, Michael Rudolph, Lisa Cavacini, Miklos Guttman, and Nicholas Mantis

Corresponding Author(s): Nicholas Mantis, Wadsworth Center, New York State Department of Health

Review Timeline:

Submission Date:	November 12, 2024
Editorial Decision:	December 9, 2024
Revision Received:	January 6, 2025
Accepted:	January 7, 2025

Editor: Catherine Brissette

Reviewer(s): Disclosure of reviewer identity is with reference to reviewer comments included in decision letter(s). The following individuals involved in review of your submission have agreed to reveal their identity: Suman Kundu (Reviewer #2)

Transaction Report:

DOI: <https://doi.org/10.1128/spectrum.02883-24>

Re: Spectrum02883-24 (A type-specific B cell epitope at the apex of Outer surface protein C (OspC) of the Lyme disease spirochete, *Borrelia burgdorferi*)

Dear Dr. Nicholas J. Mantis:

Thank you for the privilege of reviewing your work. Below you will find my comments, instructions from the Spectrum editorial office, and the reviewer comments.

While both reviewers found merit in your manuscript, modifications are needed. Please pay special attention to Major Comment #2 from reviewer 1. Also, there were numerous errors in labeling, grammar, etc. that made the review more challenging. Please pay careful attention to the minor edits and suggestions.

Revision Guidelines

Sincerely,
Catherine Brissette
Editor
Microbiology Spectrum

Reviewer #1 (Comments for the Author):

This manuscript takes a structural biology approach to understanding the type specificity of naturally arising antibodies to OspC, a dominant outer surface protein of tick-transmitted *Borrelia burgdorferi* spirochetes that is required for infection of the mammalian host.

The authors have generated a mouse mAb to rOspC type A that is specific for *Borrelia* with that Osp C type only and that interferes with spirochete motility in a complement independent fashion. Review of the literature indicates that similar mAbs have been reported in the past. These authors have gone a bit further to characterize more specifically the antigenic epitope of a humanized rOspC type A-specific mAb by amino acid substitutions and have identified critical residues that confer binding of the mAb. They have also shown the critical roles that residues E140 and D144 play in the binding of antibodies in Lyme biobank sera previously testing positive for reactivity to residues 130-150 of rOspC type A. The study is well designed and conclusions are generally supported by the data provided, with an exception perhaps being the results of the aggregation studies (see major comment 2). The manuscript could be improved by addressing issues related to consistency of nomenclature/abbreviations, figure labelling, and attention to grammar/syntax issues, as outlined in minor comments below.

Major comments

1. Introduction lines 72-74: The reference to the "Vsp family unique to the *Borrelia* and *Borrelia* genera" could be a bit confusing to some readers. First, not all readers may be aware that *Borrelia* is referring to relapsing fever spirochetes; the renaming of Lyme spirochetes as *Borrelia* has not been consistently used in the literature. Second, most Lyme references to vsps are in the context of discussions of vlsE, which undergoes antigenic variation within a single bacterial strain, although through a different method than the Vsps of relapsing fever spirochetes. Suggest eliminating mention of vsps and *Borrelia* (RF) spirochetes.
2. Results: Figure 1 and lines 104, 106-108. Line 104 refers to Figure 2 but the authors mean Figure 1. In Figure 1B, the panels for mAb 8C1 do not have the percentages of black and red dots expressed. The degree of agglutination is visually subtle in comparison to the authors' JI paper with OspA mAbs. Did they try temperature shifting the spirochetes to increase OspC expression on the cultured organisms? Alternatively, is there a reason why they didn't use the strain with the IPTG inducible rpoS allele for this experiment? Results may be more robust if the level of surface expression of OspC expression is increased.
3. Please define the control mAb PB10. Neither Figure 1 nor the methods indicate its source. Is this the humanized ricin mAb?
4. Table 1, lines 121-122. Do the authors think that a concentration of 10 µg/ml of the rOspC mAb binding this epitope can be achieved in a mammalian host in the absence of vaccination? Just wondering whether this in vitro result is physiologically relevant. The B5 mAb had an effect at 3 µg/ml. The mAb16.22 in the Yang reference also has evidence of a bactericidal effect at a much lower concentration, although they used a different assay.
5. Lines 319-320: Suggest referring to "Human sera samples testing positive for Lyme disease" instead of present wording. By what method were sera tested for Lyme disease? The biobank source should be able to give you this information. Are these samples only IgG or were IgM positive samples from people with early Lyme disease included?

Minor comments

1. Line 52: the word "of" is missing
2. Line 67: Arthritis is also a complication of Lyme disease. Consider adding "arthritic" to this list.
3. Line 96: Supernatants "contain" IgG, they don't "secrete" it.
4. Line 96: Luminex was likely conducted using recombinant Osps. Please use consistent nomenclature throughout the manuscript when referring to assays using rOsps. Leaving out the word "recombinant" or the letter "r" from the abbreviation may mislead readers into thinking that some assays are conducted with native Osps.
5. Line 106: It would be helpful to refer to Figure 1A.
6. Line 108: This should be labelled Figure 1B,C.
7. Line 165: Current convention is that disease names are not used as adjectives with patients. This should be rephrased as "people with Lyme disease" or equivalent.
8. Line 166: Human linear epitopes might be better stated as "linear epitopes antigenic in humans that are dependent on..."
9. Line 215: What *Borrelia* strain is the rOspA based on?
10. Lines 226-227: Correct grammar. Capitalize P in next sentence.
11. Line 237: Paragraphs on "Mouse immunizations and hybridoma production" and on "Direct and capture ELISAs": please be consistent with recombinant OspC abbreviation (preferably rOspC with indication as needed for type).
12. Line 271: remove "the"
13. Lines 293-296: This is not a complete sentence.
14. Line 301: "PB10" is not defined anywhere in the manuscript.
15. Line 314: Insert temperature unit

Reviewer #2 (Comments for the Author):

Vance and the team investigated the type-specific B cell epitope of the major outer surface protein C (OspC) of *Borrelia burgdorferi*, the spirochete responsible for Lyme disease. In their study, they produced and characterized a mouse monoclonal antibody, 8C1, which recognizes both native and recombinant forms of OspC type A. This antibody effectively arrested *Borrelia* motility in vitro, both with and without involvement of complement.

Through epitope mapping, the group identified the immune-dominant region of OspC as being located at the apex of alpha-helix 3 (residues 130-150). They further validated the reactivity of this immunodominant region by analyzing serum samples from 26 human patients positive for Lyme disease.

While this study is significant in confirming an immunodominant region of OspC, it falls short of advancing Lyme disease vaccine design, leaving room for further research and refinement.

Below are the recommendations to improve the manuscript quality:

1. The author mentioned *B. burgdorferi* strain B31 which is known to have OspC type A antigen. Please clarify and specify the relevance of the *B. burgdorferi* strain B313 and B31A3 to the study, particularly in relation to Osp expression.
2. Are the immunodominant regions of OspC conserved or variable? If they are variable, what advantage does producing a monoclonal antibody against a single epitope provide, given that it binds to a specific OspC type? Additionally, how can this be considered a viable approach for vaccine development? For vaccine development, focusing on conserved epitopes, or employing a multivalent approach that includes multiple OspC variants, may provide broader protection against diverse *Borrelia burgdorferi* strains.
3. What strain of *Borrelia burgdorferi* was used for the in vitro motility arrest assay? Please provide detailed information on *B. burgdorferi* strain GGW941. Additionally, why was the B31 strain not used to confirm the type-specific reactivity?
4. Authors are encouraged to include few references that provides details of OspC variants found in the infected *Ixodes* nymphal ticks and OspC derived protection (PMID: 38360853, 31959423, 39221484).
5. Do *Borrelia* express OspC in culture with the same heterogeneity observed during infection? What are the OspC types dominant in different host such as mouse, human and ticks? Please include a section either in the introduction or in the discussion.
6. Beside the flowcytometric validation did the author checked surface labelling to *B. burgdorferi* by fluorescent or confocal microscopy to validate the specific binding to the antigen OspC?
7. Did the author check the protection against *B. burgdorferi* after immunization in mice in the same way they produced the monoclonal antibody?
8. It would be interesting to see the protective efficacy of the antibody 8C1 in vivo through a OspC type-specific tick challenge. The manuscript can be accepted after recommended revisions are made.

This manuscript takes a structural biology approach to understanding the type specificity of naturally arising antibodies to OspC, a dominant outer surface protein of tick-transmitted *Borrelia* spirochetes that is required for infection of the mammalian host. The authors have generated a mouse mAb to rOspC type A that is specific for *Borrelia* with that Osp C type only and that interferes with spirochete motility in a complement independent fashion. Review of the literature indicates that similar mAbs have been reported in the past. These authors have gone a bit further to characterize more specifically the antigenic epitope of a humanized rOspC type A-specific mAb by amino acid substitutions and have identified critical residues that confer binding of the mAb. They have also shown the critical roles that residues E140 and D144 play in the binding of antibodies in Lyme biobank sera previously testing positive for reactivity to residues 130-150 of rOspC type A. The study is well designed and conclusions are generally supported by the data provided, with an exception perhaps being the results of the aggregation studies (see major comment 2). The manuscript could be improved by addressing issues related to consistency of nomenclature/abbreviations, figure labelling, and attention to grammar/syntax issues, as outlined in minor comments below.

Major comments

1. Introduction lines 72-74: The reference to the "Vsp family unique to the *Borrelia* and *Borrelia* genera" could be a bit confusing to some readers. First, not all readers may be aware that *Borrelia* is referring to relapsing fever spirochetes; the renaming of Lyme spirochetes as *Borrelia* has not been consistently used in the literature. Second, most Lyme references to vsps are in the context of discussions of vlsE, which undergoes antigenic variation within a single bacterial strain, although through a different method than the Vsps of relapsing fever spirochetes. Suggest eliminating mention of vsps and *Borrelia* (RF) spirochetes.
2. Results: Figure 1 and lines 104, 106-108. Line 104 refers to Figure 2 but the authors mean Figure 1. In Figure 1B, the panels for mAb 8C1 do not have the percentages of black and red dots expressed. The degree of agglutination is visually subtle in comparison to the authors' JI paper with OspA mAbs. Did they try temperature shifting the spirochetes to increase OspC expression on the cultured organisms? Alternatively, is there a reason why they didn't use the strain with the IPTG inducible rpoS allele for this experiment? Results may be more robust if the level of surface expression of OspC expression is increased.
3. Please define the control mAb PB10. Neither Figure 1 nor the methods indicate its source. Is this the humanized ricin mAb?
4. Table 1, lines 121-122. Do the authors think that a concentration of 10 µg/ml of the rOspC mAb binding this epitope can be achieved in a mammalian host in the absence of vaccination? Just wondering whether this in vitro result is physiologically relevant. The B5 mAb had an effect at 3 µg/ml. The mAb16.22 in the Yang reference also has evidence of a bactericidal effect at a much lower concentration, although they used a different assay.

5. Lines 319-320: Suggest referring to "Human sera samples testing positive for Lyme disease" instead of present wording. By what method were sera tested for Lyme disease? The biobank source should be able to give you this information. Are these samples only IgG or were IgM positive samples from people with early Lyme disease included?

Minor comments

1. Line 52: the word "of" is missing
2. Line 67: Arthritis is also a complication of Lyme disease. Consider adding "arthritic" to this list.
3. Line 96: Supernatants "contain" IgG, they don't "secrete" it.
4. Line 96: Luminex was likely conducted using recombinant Osps. Please use consistent nomenclature throughout the manuscript when referring to assays using rOsps. Leaving out the word "recombinant" or the letter "r" from the abbreviation may mislead readers into thinking that some assays are conducted with native Osps.
5. Line 106: It would be helpful to refer to Figure 1A.
6. Line 108: This should be labelled Figure 1B,C.
7. Line 165: Current convention is that disease names are not used as adjectives with patients. This should be rephrased as "people with Lyme disease" or equivalent.
8. Line 166: Human linear epitopes might be better stated as "linear epitopes antigenic in humans that are dependent on..."
9. Line 215: What Borrelia strain is the rOspA based on?
10. Lines 226-227: Correct grammar. Capitalize P in next sentence.
11. Line 237: Paragraphs on "Mouse immunizations and hybridoma production" and on "Direct and capture ELISAs": please be consistent with recombinant OspC abbreviation (preferably rOspC with indication as needed for type).
12. Line 271: remove "the"
13. Lines 293-296: This is not a complete sentence.
14. Line 301: "PB10" is not defined anywhere in the manuscript.
15. Line 314: Insert temperature unit

Reviewer #1 (Comments for the Author):

This manuscript takes a structural biology approach to understanding the type specificity of naturally arising antibodies to OspC, a dominant outer surface protein of tick-transmitted *Borrelia* spirochetes that is required for infection of the mammalian host. The authors have generated a mouse mAb to rOspC type A that is specific for *Borrelia* with that Osp C type only and that interferes with spirochete motility in a complement independent fashion. Review of the literature indicates that similar mAbs have been reported in the past. These authors have gone a bit further to characterize more specifically the antigenic epitope of a humanized rOspC type A-specific mAb by amino acid substitutions and have identified critical residues that confer binding of the mAb. They have also shown the critical roles that residues E140 and D144 play in the binding of antibodies in Lyme biobank sera previously testing positive for reactivity to residues 130-150 of rOspC type A. The study is well designed and conclusions are generally supported by the data provided, with an exception perhaps being the results of the aggregation studies (see major comment 2). The manuscript could be improved by addressing issues related to consistency of nomenclature/abbreviations, figure labelling, and attention to grammar/syntax issues, as outlined in minor comments below.

Major comments

1. Introduction lines 72-74: The reference to the "Vsp family unique to the *Borrelia* and *Borrelia* genera" could be a bit confusing to some readers. First, not all readers may be aware that *Borrelia* is referring to relapsing fever spirochetes; the renaming of Lyme spirochetes as *Borrelia* has not been consistently used in the literature. Second, most Lyme references to vsps are in the context of discussions of vlsE, which undergoes antigenic variation within a single bacterial strain, although through a different method than the Vsps of relapsing fever spirochetes. Suggest eliminating mention of vsps and *Borrelia* (RF) spirochetes. Response: The Reviewer's comments are acknowledged. As suggested, we have removed mention of relapsing fever (RF) and related Vsps.

2. Results: Figure 1 and lines 104, 106-108. Line 104 refers to Figure 2 but the authors mean Figure 1. In Figure 1B, the panels for mAb 8C1 do not have the percentages of black and red dots expressed. The degree of agglutination is visually subtle in comparison to the authors' JI paper with OspA mAbs. Did they try temperature shifting the spirochetes to increase OspC expression on the cultured organisms? Alternatively, is there a reason why they didn't use the strain with the IPTG inducible rpoS allele for this experiment? Results may be more robust if the level of surface expression of OspC expression is increased. Response: We thank Reviewer 1 for noting the incorrect Figure citation, as well as the lack of annotation of agglutination. Both have been corrected.

Regarding the specifics of agglutination, identifying ideal OspC expression conditions *in vitro* has proven vexing. In the current study, the methods and *B. burgdorferi* strains are identical to those reported by Rudolph 2023 (mBio) and 2024 (Journal of Immunology). Namely, the spirochetes were cultured at 37°C to mid log phase. The difference in agglutination across the different OspC MAbs (B5, B11, 8C1) is likely related to both antibody avidity and epitope specificity. We have inserted text in the Results to underscore this point.

Finally, we have historically relied on the B313 strain as our OspC_A positive control for our flow cytometry studies alongside 297 and ZS7 under similar growth conditions. We are currently developing *rpoS* overexpression strain for each to enable side-by-side comparisons.

3. Please define the control mAb PB10. Neither Figure 1 nor the methods indicate its source. Is this the humanized ricin mAb?

Response: We apologize for the oversight. PB10 is ricin toxin-specific antibody that serves as an IgG1 isotype control. We have noted this in the text and the legend to Figure 1. We have included a PB10 citation.

4. Table 1, lines 121-122. Do the authors think that a concentration of 10 µg/ml of the rOspC mAb binding this epitope can be achieved in a mammalian host in the absence of vaccination? Just wondering whether this in vitro result is physiologically relevant. The B5 mAb had an effect at 3 µg/ml. The mAb16.22 in the Yang reference also has evidence of a bactericidal effect at a much lower concentration, although they used a different assay.

Response: Indeed, the Reviewer makes a valid point about the translatable nature of the *in vitro* results. In fact, in preliminary passive transfer studies, 8C1 has not shown any protective efficacy against intradermal challenge model, which we attribute to 8C1's rather low apparent affinity (11 nM) compared to B5 (10 pM) for example. We have added commentary to this effect into the manuscript: "The relevance of these observations to *in vivo* activity remains to be determined, as the exact concentrations and mechanism(s) by which OspC antibodies interfere with *B. burgdorferi* transmission and infection remain incompletely defined...."

5. Lines 319-320: Suggest referring to "Human sera samples testing positive for Lyme disease" instead of present wording. By what method were sera tested for Lyme disease? The biobank source should be able to give you this information. Are these samples only IgG or were IgM positive samples from people with early Lyme disease included?

Response: The text has been modified to state: We employed human serum samples provided by the Lyme Disease Biobank at Nuvance Health (Danbury, CT) that had been deemed positive in a standard or modified two-tiered Lyme disease test. As samples were principally from individuals with post-treatment Lyme disease (PTLD), only IgG reactivity was examined.

Minor comments

1. Line 52: the word "of" is missing. Response: Corrected.

2. Line 67: Arthritis is also a complication of Lyme disease. Consider adding "arthritic" to this list. Response: Corrected.

3. Line 96: Supernatants "contain" IgG, they don't "secrete" it. Response: Corrected.

4. Line 96: Luminex was likely conducted using recombinant Osps. Please use consistent nomenclature throughout the manuscript when referring to assays using rOsps. Leaving out the word "recombinant" or the letter "r" from the abbreviation may mislead readers

into thinking that some assays are conducted with native Osps. Response: Corrected as recommended.

5. Line 106: It would be helpful to refer to Figure 1A. Response: Corrected.

6. Line 108: This should be labelled Figure 1B,C. Response: Corrected.

7. Line 165: Current convention is that disease names are not used as adjectives with patients. This should be rephrased as "people with Lyme disease" or equivalent. Response: Corrected.

8. Line 166: Human linear epitopes might be better stated as "linear epitopes antigenic in humans that are dependent on..." Response: Changed.

9. Line 215: What Borreliella strain is the rOspA based on? Response: rOspA was based on strain B31. This is now noted in the methods along with the UniProtIDs for each OspA type.

10. Lines 226-227: Correct grammar. Capitalize P in next sentence. Response: Corrected.

11. Line 237: Paragraphs on "Mouse immunizations and hybridoma production" and on "Direct and capture ELISAs": please be consistent with recombinant OspC abbreviation (preferably rOspC with indication as needed for type). Response: Fixed.

12. Line 271: remove "the" Response: Removed.

13. Lines 293-296: This is not a complete sentence. Response: Corrected.

14. Line 301: "PB10" is not defined anywhere in the manuscript. Response: Corrected, as noted above.

15. Line 314: Insert temperature unit. Response: Corrected.

Reviewer #2 (Comments for the Author):

Vance and the team investigated the type-specific B cell epitope of the major outer surface protein C (OspC) of *Borrelia burgdorferi*, the spirochete responsible for Lyme disease. In their study, they produced and characterized a mouse monoclonal antibody, 8C1, which recognizes both native and recombinant forms of OspC type A. This antibody effectively arrested *Borrelia* motility in vitro, both with and without involvement of complement. Through epitope mapping, the group identified the immune-dominant region of OspC as being located at the apex of alpha-helix 3 (residues 130-150). They further validated the reactivity of this immunodominant region by analyzing serum samples from 26 human patients positive for Lyme disease. While this study is significant in confirming an immunodominant region of OspC, it falls short of advancing Lyme disease vaccine design, leaving room for further research and refinement.

Below are the recommendations to improve the manuscript quality:

1. The author mentioned *B. burgdorferi* strain B31 which is known to have OspC type A antigen. Please clarify and specify the relevance of the *B. burgdorferi* strain B313 and B31A3 to the study, particularly in relation to Osp expression. Response: We have now provided a strain list in Table 1 with relevant citations and notes about OspC phenotype. Included in the table are citations for strain B31-A3 (Infect Immun. 2002 Apr;70(4):2139-50) and B313 (Infect Immun. 1995 Apr;63(4):1573-80).

2. Are the immunodominant regions of OspC conserved or variable? If they are variable, what advantage does producing a monoclonal antibody against a single epitope provide, given that it binds to a specific OspC type? Additionally, how can this be considered a viable approach for vaccine development? For vaccine development, focusing on conserved epitopes, or employing a multivalent approach that includes multiple OspC variants, may provide broader protection against diverse *Borrelia burgdorferi* strains. Response: Reviewer 2 raises the primary challenges associated with OspC-based vaccines for Lyme disease. At present, a comprehensive B cell epitope map of OspC is lacking, so it is not known which epitopes are conserved and which are variable. Our goal is to identify both the variable and conserved B cell epitopes on OspC and use this information to for the sake of vaccine design, to retain the conserved epitopes and “engineer out” the type-specific epitopes.

3. What strain of *Borrelia burgdorferi* was used for the in vitro motility arrest assay? Please provide detailed information on *B. burgdorferi* strain GGW941. Additionally, why was the B31 strain not used to confirm the type-specific reactivity? Response: B31 was used for the motility arrest studies. As noted above, B31 tends to express low levels of OspC (type A) when grown in culture. For that reason, we opted to activate ospC gene expression by providing RpoS in trans in the B31 background. As such, we are using B31 to confirm type specificity of 8C1. As requested, we have now provided detailed information on strain GGW941 in Table 1. Strain B313 expresses high level of OspC but is intrinsically serum sensitive, which precludes its use in functional assays that involve exogenous complement.

4. Authors are encouraged to include few references that provides details of OspC variants found in the infected *Ixodes* nymphal ticks and OspC derived protection (PMID: 38360853, 31959423, 39221484).

Response: The Reviewer makes an important point. As requested, we have modified the text to describe OspC types associated with human disease and have inserted citations indicating that OspC vaccination elicits protective immunity. For example, the text state that “In New York state, for example, OspC types A, B and K, which are associated with more invasive disease, represent ~70% of all isolates”.

We also inserted this text in the Introduction (see manuscript for references):

“Even within relatively confined geographical areas, the diversity of ospC alleles within tick-associated *B. burgdorferi* can be remarkably high. As a case in point, 19 different ospC alleles were identified within a survey of nymphal and adult ticks (*Ixodes scapularis*) from a region of

high endemicity in New York State. Others have noted similar degrees of ospC diversity within colonies of wild caught ticks.

And "...The polymorphic nature of ospC represents one of the major challenges associated with the use of OspC as a candidate Lyme vaccine. Marconi and colleagues have overcome the challenge by generating "chimeritope" antigens consisting of concatenated epitopes (polypeptides) encompassing multiple OspC types. However, the "chimeritope" approach does not retain conformational tertiary and quaternary epitopes, including some associated with protection {Rudolph, 2023 #7644}. An alternative approach is to preserve OspC's quaternary structure but "engineer out" immunodominant variable epitopes. This strategy is referred to as immune focusing or protein dissection and has been applied widely to other variable antigens of interest, like HIV-1's surface glycoprotein. With that in mind, we have sought to better define the antigenic landscape of OspC and generate a high-resolution B cell epitope map of the molecule as a basis for rational vaccine design."

Citations now include:

Izac JR, O'Bier NS, Oliver LD Jr, Camire AC, Earnhart CG, LeBlanc Rhodes DV, Young BF, Parnham SR, Davies C, Marconi RT. Development and optimization of OspC chimeritope vaccinogens for Lyme disease. *Vaccine*. 2020 Feb 18;38(8):1915-1924. doi: 10.1016/j.vaccine.2020.01.027. Epub 2020 Jan 17. PMID: 31959423; PMCID: PMC7085410.

Gingerich MC, Nair N, Azevedo JF, Samanta K, Kundu S, He B, Gomes-Solecki M. Intranasal vaccine for Lyme disease provides protection against tick transmitted *Borrelia burgdorferi* beyond one year. *NPJ Vaccines*. 2024 Feb 15;9(1):33. doi: 10.1038/s41541-023-00802-y. PMID: 38360853; PMCID: PMC10869809.

Di L, Wan Z, Akther S, Ying C, Larracuent A, Li L, Di C, Nunez R, Cucura DM, Goddard NL, Krampis K, Qiu WG. Genotyping and Quantifying Lyme Pathogen Strains by Deep Sequencing of the Outer Surface Protein C (*ospC*) Locus. *J Clin Microbiol*. 2018 Oct 25;56(11):e00940-18. doi: 10.1128/JCM.00940-18. PMID: 30158192; PMCID: PMC6204676.

5. Do *Borrelia* express OspC in culture with the same heterogeneity observed during infection? What are the OspC types dominant in different host such as mouse, human and ticks? Please include a section either in the introduction or in the discussion.

Response: This is an extremely broad area of research that only peripherally relates to our study focused on one particular monoclonal antibody. However, as noted above, we have included citations related to different OspC types in both tick and human isolates.

6. Beside the flowcytometric validation did the author checked surface labelling to *B. burgdorferi* by fluorescent or confocal microscopy to validate the specific binding to the antigen OspC? Response: We have confirmed OspC reactivity by whole *B. burgdorferi* dot blot along with quantitative flow cytometry but are only now in the process of optimizing immunostaining of live spirochetes for confocal microscopy. We hesitate to perform staining on fixed (non-viable cells) due to possible effects on membrane permeability.

7. Did the author check the protection against *B. burgdorferi* after immunization in mice in the same way they produced the monoclonal antibody? Response: The Reviewer raises an excellent point that we have not addressed experimentally because at the time we generated 8C1 we had not reproducibly established *B. burgdorferi* challenge protocols. However, those studies are ongoing with a focus on human and mouse derived OspC MAbs.

8. It would be interesting to see the protective efficacy of the antibody 8C1 in vivo through a OspC type-specific tick challenge. Response: Indeed, those studies are ongoing in needle injection model as well as tick challenge model. In preliminary passive transfer studies, 8C1 has not shown any protective efficacy against intradermal challenge model. Final results will be reported in a separate manuscript alongside B5 and B11.

The manuscript can be accepted after recommended revisions are made.

Re: Spectrum02883-24R1 (A type-specific B cell epitope at the apex of Outer surface protein C (OspC) of the Lyme disease spirochete, *Borrelia burgdorferi*)

Dear Dr. Nicholas J. Mantis:

Your manuscript has been accepted, and I am forwarding it to the ASM production staff for publication. Your paper will first be checked to make sure all elements meet the technical requirements. ASM staff will contact you if anything needs to be revised before copyediting and production can begin. Otherwise, you will be notified when your proofs are ready to be viewed.

Sincerely,
Catherine Brissette
Editor
Microbiology Spectrum